# Comparison of long COVID, recovered COVID, and non-COVID Post-Acute Infection Syndromes over three years

Caleb R. Carr[1], Nicole L. Gentile[2,3], Jeanne Bertolli[4], Warren Szewczyk[5], Jin-Mann S. Lin[4], Elizabeth R. Unger[4], Quan M. Vu[4], Nona Sotoodehnia[6], Annette L. Fitzpatrick[2,5]*

1 University of Washington School of Medicine, Seattle, Washington, United States of America, 2 Department of Family Medicine, University of Washington, Seattle, Washington, United States of America, 3 Department of Laboratory Medicine and Pathology, University of Washington, Seattle, Washington, United States of America, 4 Division of High-Consequence Pathogens and Pathology, National Center for Emerging and Zoonotic Infectious Diseases, Centers for Disease Control and Prevention, Atlanta, Georgia, United States of America, 5 Department of Epidemiology, University of Washington, Seattle, Washington, United States of America, 6 Division of Cardiology, Department of Medicine, University of Washington, Seattle, Washington, United States of America

* fitzpal@uw.edu

## Abstract

### Background

Comparing the characteristics of patients with long COVID to those with other post-acute infection syndromes (PAIS) could potentially provide clues to common underlying disease processes that may affect patient recovery.

### Methods

We identified records of patients who had documented SARS-CoV-2 tests in the University of Washington Medicine electronic health record (EHR) database from January 1, 2019, through January 31, 2022 (n = 139,472). Patients were classified into three groups: 1) long COVID defined by a positive SARS-CoV-2 test and a long COVID-related diagnosis code (n = 580); 2) recovered COVID defined by a positive test and no long COVID associated diagnosis codes (n = 7,437); and 3) non-COVID PAIS defined by a negative test, non-SARS-CoV-2 related PAIS diagnosis codes, and no COVID related codes (n = 106). Using multivariate logistic regression, we compared the clinical characteristics of these groups at three timeframes to address preclinical, acute and post-acute diagnoses: before index SARS-CoV-2 test, within 30 days of index test, and > 30 days after index test.

**Data availability statement:** Our data have ethical restrictions for sharing as the source is from electronic health records of patients in a healthcare system whose consent have been waived. Although medical record numbers have been removed, other data including dates and locations of care are included. It may be possible to identify individuals from one or a combination of data fields. Our IRB approval through the University of Washington does not allow our sharing of these records. Should someone wish to use our data, we can accommodate requests by addressing their individual needs. Data requests may be made to: Dr. Kari Stephens Vice Chair for Research Department of Family Medicine University of Washington, Seattle WA 98195 kstephen@uw.edu

**Funding:** Funding for this project was received through a contract from the Centers for Disease Control and Prevention (CDC), contract number 75D30121C10207. The funder's role was consultation on the analysis and interpretation of data, providing input on the decision to publish, and participation in the preparation of the manuscript.

**Competing interests:** The authors have declared that no competing interests exist.

**Abbreviations:** SARS-CoV-2, severe acute respiratory syndrome coronavirus 2; COVID-19, coronavirus disease 2019; PCR, polymerase chain reaction; AIS, post-acute infection syndrome; EHR, electronic health records; IQR, interquartile rangeCCI, Charlson comorbidity index; ICD-10-CM, international classification of diseases, 10th revision, clinical modification.

## Results

The long COVID group had a higher Charlson comorbidity index [median (IQR), 2 (0–4)] than the other two patient groups [median (IQR), 1 (0–3) and 1 (0–3)]. The long COVID and non-COVID PAIS patients were older and had greater smoking exposure than the recovered COVID group. Compared to the recovered COVID control group, the long COVID group had more health problems prior to the infection, including respiratory and metabolic as well as more severe infections and comorbidities based on the ICD codes found in the acute phase records. In the post-acute timeframe, many symptoms were more likely to be associated with long COVID than recovered patients with COVID-19 including abnormalities of heart beat [OR (95% CI), 5.31 (3.96–7.13)], cognition, perception, or emotional state symptoms [OR (95% CI), 5.14 (3.81–6.92)], malaise and fatigue [OR (95% CI), 4.20 (3.13–5.63)], and sleep disorders [OR (95% CI), 2.47, (1.79–3.43)], all $p < 0.05$. In contrast, the non-COVID PAIS group shared many similarities with the long COVID group across all three timeframes.

## Conclusions

Patients diagnosed with long COVID were more similar to patients with a non-COVID-related PAIS than to recovered patients with COVID-19. This suggests risk factors for PAIS may be similar and independent of the infectious agent.

---

## Introduction

Severe acute respiratory syndrome coronavirus 2 (SARS-CoV-2) is the cause of coronavirus disease 2019 (COVID-19), which has devastated healthcare systems worldwide with hundreds of millions of confirmed infections and millions of deaths [1]. During the early stages of the COVID-19 pandemic, it was reported that individuals were having persistent or developing new symptoms following a SARS-CoV-2 infection [2,3], a post-acute infection syndrome (PAIS) now known as long COVID [4]. While many patients recover from SARS-CoV-2 infection within a few months, data from the 2022 National Health Interview Survey estimated that almost 7% of adults, equivalent to about 18 million Americans, reported having long-term symptoms, or long COVID, following the acute phase of the disease [5].

The observation of unexplained chronic sequelae following an acute infection is not a new phenomenon. PAISs have previously been documented following other infections caused by viruses, bacteria, and parasites, including Ebola, dengue, polio, chikungunya, Epstein-Barr virus and other SARS infections [6–11]. However, many infections are not diagnosed during the acute illness, and even for diagnosed infections, post-acute symptoms often occur much later, making them challenging to link to the initial infection [12]. In addition, PAIS often go undiagnosed due to the non-specific nature of the symptoms and the lack of a diagnostic biomarker(s) [6]. However, the large numbers of people experiencing long-term negative health impacts following COVID-19 has brought more attention to PAIS.

As recent studies have shown, many of the symptoms associated with long COVID are common in general populations [13,14]. Several studies have used control cohorts that were infected with different pathogens, such as influenza, to help elucidate features specific to long COVID vs other PAIS [15–19]. These studies have contributed to a growing body of evidence indicating that similar long-term symptom burdens follow a variety of different infections, and that COVID-19 is no more likely than other acute infections to be associated with myalgic encephalomyelitis/chronic fatigue syndrome, a debilitating condition often preceded by an infection-like illness [20,21]. Questions remain about why some people with these infections recover from the acute infection, and others have long-term debilitating sequelae.

This study leverages electronic health record (EHR) data from a large healthcare system with some of the earliest COVID-19 pandemic data in the United States to provide an extensive longitudinal timeframe for analysis. To identify factors associated with recovery versus long-term symptoms, we compared individuals with EHR evidence of long COVID to those who recovered from the acute phase of COVID and to others who were not infected by SARS-CoV-2 (did not have a COVID-19 diagnosis code) but were diagnosed with a post-acute infection syndrome, presumed to be from a non-SARS-CoV-2 pathogen (non-COVID PAIS). Data, including diagnoses and symptoms coded in the medical record before, during and more than 30 days after the index SARS-CoV-2 test, available demographics, and specific healthcare factors, were reviewed to identify similarities and differences between long COVID, recovered COVID, and non-COVID PAIS.

## Methods

### EHR database

We identified patients in the University of Washington Medicine (Seattle, WA) EHR database who received a SARS-CoV-2 PCR test from January 1, 2019, through January 31, 2022. This cohort included all individuals with positive (n = 9,408) and negative (n = 130,064) results during this timeframe who were ≥18 years old and had a medical encounter with an International Classification of Diseases, 10th Revision, Clinical Modification (ICD-10-CM) diagnostic code on record. The total sample of 139,472 patients excluded individuals who received a PCR test through UW Medicine but did not receive medical care inside the UW Medicine network following the test.

All EHR data associated with the cohort were de-identified and provided to researchers with a unique identification number to link patient encounters from different medical record data tables. A waiver of informed consent was approved by the University of Washington institutional review board (IRB), Division of Human Subjects (STUDY00010039). This activity was reviewed by the Centers for Disease Control and Prevention (CDC) and was conducted consistent with applicable federal law and CDC policy. This research was undertaken as part of the Research on COVID-19 Long-term Illness, Effects, and Risk Factors (COVID-RELIEF) study at the University of Washington.

### Selection of patient groups and timeframes

We used 30 days (about 4 weeks) from the index PCR test date as the cut point to define the post-acute infection period because previous literature indicated many symptoms associated with post-COVID-19 conditions persist or begin four weeks after the initial SARS-CoV-2 infection and early long COVID definitions used during the time of data collection utilized a four week post-infection cutoff [3,22,23]. A probable long COVID group was extracted from the SARS-CoV-2 PCR positive base population by selecting individuals with an ICD-10-CM diagnostic code of U09.9 (post COVID-19 condition, unspecified), B94.8 (sequelae of other specified infectious and parasitic diseases), and/or G93.3 (postviral fatigue syndrome) at least 30 days after their positive index SARS-CoV-2 PCR test. Because the U09.9 code was not introduced until October 1st, 2021, patients who received a U07.1 (COVID-19) code through September 30, 2021, at a healthcare encounter both prior to and 30 days after their positive index PCR test were included in this long COVID group (n = 580) (Fig 1). These ICD-10-CM codes were selected based on the recommendations of clinicians working in a long COVID clinic at the University of Washington. Note that this cohort cannot be diagnosed as true positives for long COVID based

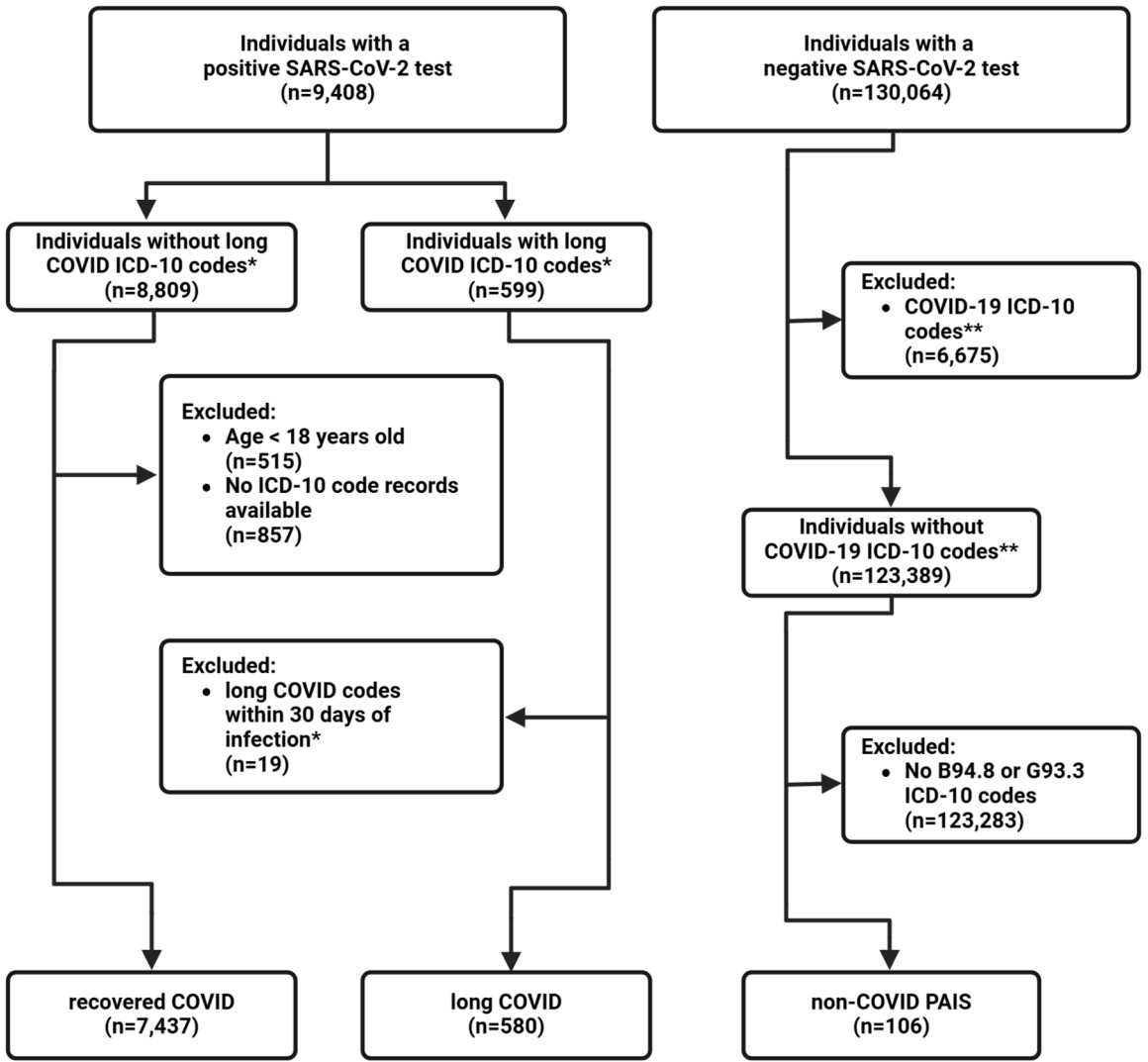

**Fig 1. Patient group selection flow diagram.** Patients were selected from the University of Washington EHR database. *: long COVID ICD-10-CM codes include U09.9, B94.8, G93.3, and U07.1 if before October 1st, 2021. **: COVID-19 related ICD-10-CM codes include U07.1, Z86.16, J12.82, U09.9, B97.21, B97.29, and B34.2.

on the current definition of symptoms lasting for at least 90 days after the initial infection but constitutes a probable long COVID positive cohort. For simplicity, long COVID will be used to describe this cohort

The remaining SARS-CoV-2 PCR-positive individuals who did not have one of the above ICD-10-CM codes were assigned to the recovered COVID group (n = 7,437). A third group was identified from the SARS-CoV-2-PCR negative group who had no indication of COVID-19 but had a post infectious syndrome ICD-10-CM code (B94.8, sequelae of other specified infectious and parasitic diseases; G93.3: postviral fatigue syndrome) (n = 106, Fig 1). No indication of COVID-19 was defined by having only negative PCR tests on record and no COVID-19-associated ICD-10-CM codes (U07.1: COVID-19; Z86.16: personal history of COVID-19; J12.82: pneumonia due to coronavirus disease 2019; U09.9: post COVID-19 condition, unspecified; B97.21: SARS-associated coronavirus as the cause of diseases classified elsewhere; B97.29: other coronavirus as the cause of diseases classified elsewhere; B34.2: coronavirus infection, unspecified).

The final samples for the three groups were: long COVID (n = 580), recovered COVID (n = 7,437) and non-COVID PAIS (n = 106).

Three timeframes with respect to the patient's index PCR test were selected to categorize the diagnoses as pre-COVID, during the COVID acute phase, and post-COVID as follows: 1) prior to the index SARS-CoV-2 PCR test (i.e., to identify pre-existing conditions); 2) within 30 days of the index SARS-CoV-2 PCR test; and 3) greater than 30 days after the index SARS-CoV-2 PCR test) and are shown in Fig 2A. For those with more than one positive test, the first positive was defined as the index infection. For consistency, the first negative SARS-CoV-2 PCR test on record was used to classify data for the non-COVID PAIS group into intervals that match those of the long COVID and recovered COVID groups. However, the infectious agent and date of index infection that led to the post-acute infection syndrome for these individuals are unknown (Fig 2).

## Data

We extracted patient demographics and ICD-10-CM codes for all healthcare encounters occurring from January 1, 2019, to January 31, 2022. Demographic data included age (in years), sex (self-reported male, female, other), race/ethnicity (self-reported White non-Hispanic, Black or African American non-Hispanic, Hispanic, Asian, American Indian or Alaska Native, Native Hawaiian or Pacific Islander, Other, Unknown), smoking exposure (current and former smokers were grouped as "yes" for smoking exposure), and hospitalization status. Hospitalization was defined as a hospital encounter within seven days of the index PCR test. Diagnostic codes were extracted from all healthcare encounters and filtered to remove any ICD-9 codes. The ICD-10 codes present prior to the index PCR test were used to calculate Charlson comorbidity index (CCI) scores indicating prevalent diseases (S1 Table) [25]. To focus on onset of new symptoms in the post-infection periods, the ICD-10 codes present prior to the index PCR test were filtered from the post-infection codes to remove pre-existing conditions/symptoms (excluding ICD-10 codes described above used to define the three patient groups). All ICD-10 codes associated with long COVID were grouped to facilitate interpretation and for presentation in figures as provided in S2 Table [3,17,23,26].

## Statistical analysis

Age and CCI scores were summarized by median and interquartile range (IQR). Categorical variables, such as sex, race/ethnicity, smoking exposure, and hospitalization, were summarized by frequency counts and percentages. For group comparisons, the two-sided Mann-Whitney $U$ test was used for continuous variables and chi-square tests were used for categorical variables. Comparison of documented symptoms/diagnoses among those with long COVID versus those in the recovered COVID and non-COVID PAIS groups was assessed using multivariable logistic regression. Separate models were run for each timeframe that identified the presence of each ICD-10 diagnostic code as a binary outcome (dependent variable), with group status as the binary indicator (independent variable), i.e., long COVID (1) or the comparison group (0), adjusted for age, sex, race/ethnicity, smoking exposure, and CCI scores for each timeframe. Adjustment variables were selected based on availability of data in the EHR and the literature. Number of ICD-codes in a patient's record was also evaluated to reflect severity of disease [27–30]. Results of analyses for each timeframe are shown separately in Fig 3 (pre-COVID), Fig 4 (acute phase of COVID) and Fig 5 (post-COVID). Analyses were conducted using Python statsmodels v0.14.1 [31]. Corrections for multiple comparisons were not done due to the exploratory nature of these analyses.

## Results

### Characteristics of patient groups

As shown in Table 1, the long COVID and non-COVID PAIS groups had similar age distributions [median (IQR), 54 (39–66) and 51 (34–63), $p = 0.12$] and were older than the recovered COVID group [median (IQR), 46 (31–61), $p < 0.001$].

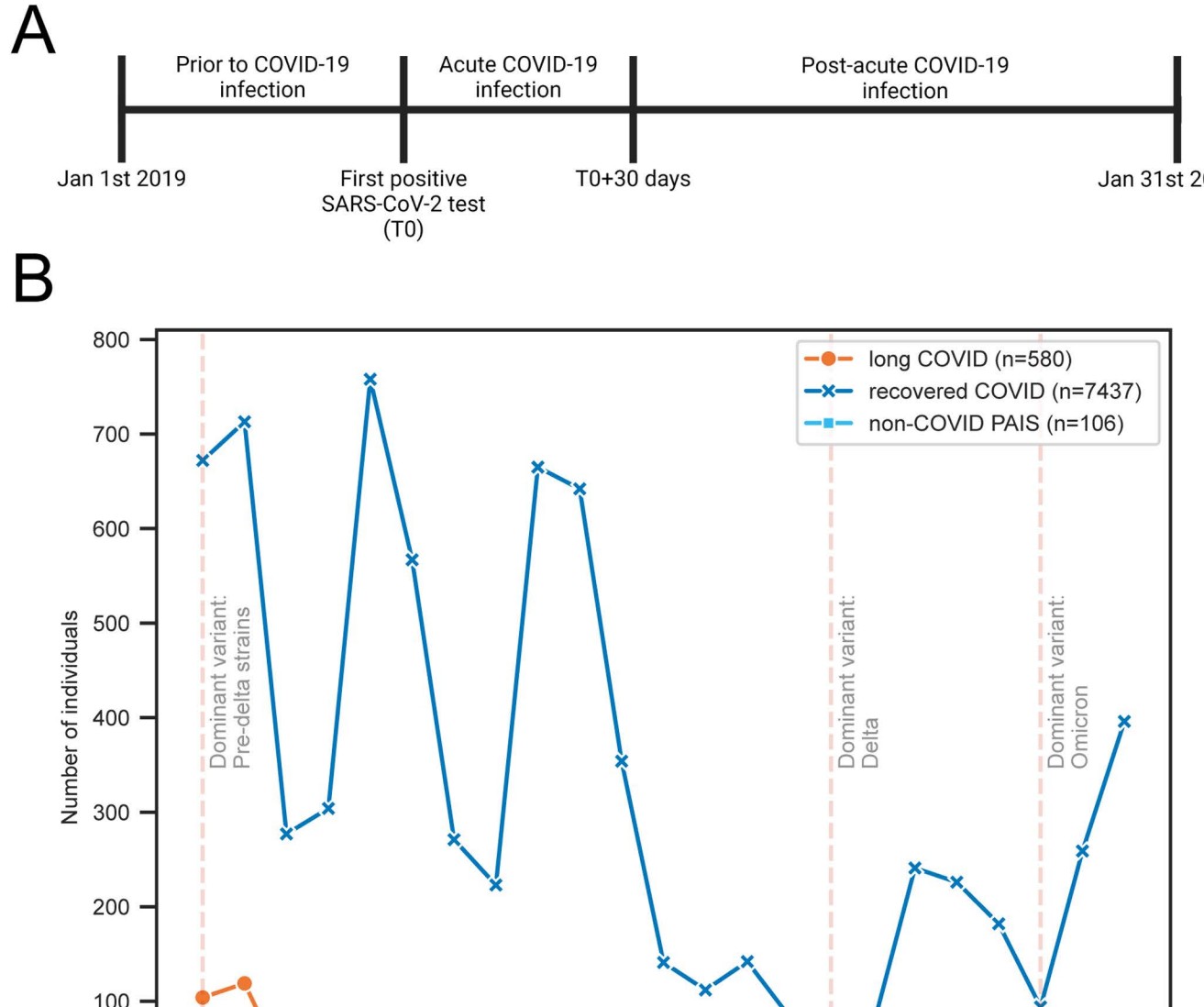

**Fig 2. Temporal views of data included for model analysis showing when individuals were tested for SARS-CoV-2 infection. (A)** Data were classified as either prior to first SARS-CoV-2 test, within 30 days of first SARS-CoV-2 test, or at least 30 days after first SARS-CoV-2 test. The first positive test was used for all individuals in the long COVID and recovered COVID cohorts while the first test on record was used for the non-COVID PAIS cohort. **(B)** The date of the SARS-CoV-2 test for each individual was binned into months for the three cohorts. The dominant variant circulating at the time in Washington State is marked by vertical dashed lines [24].

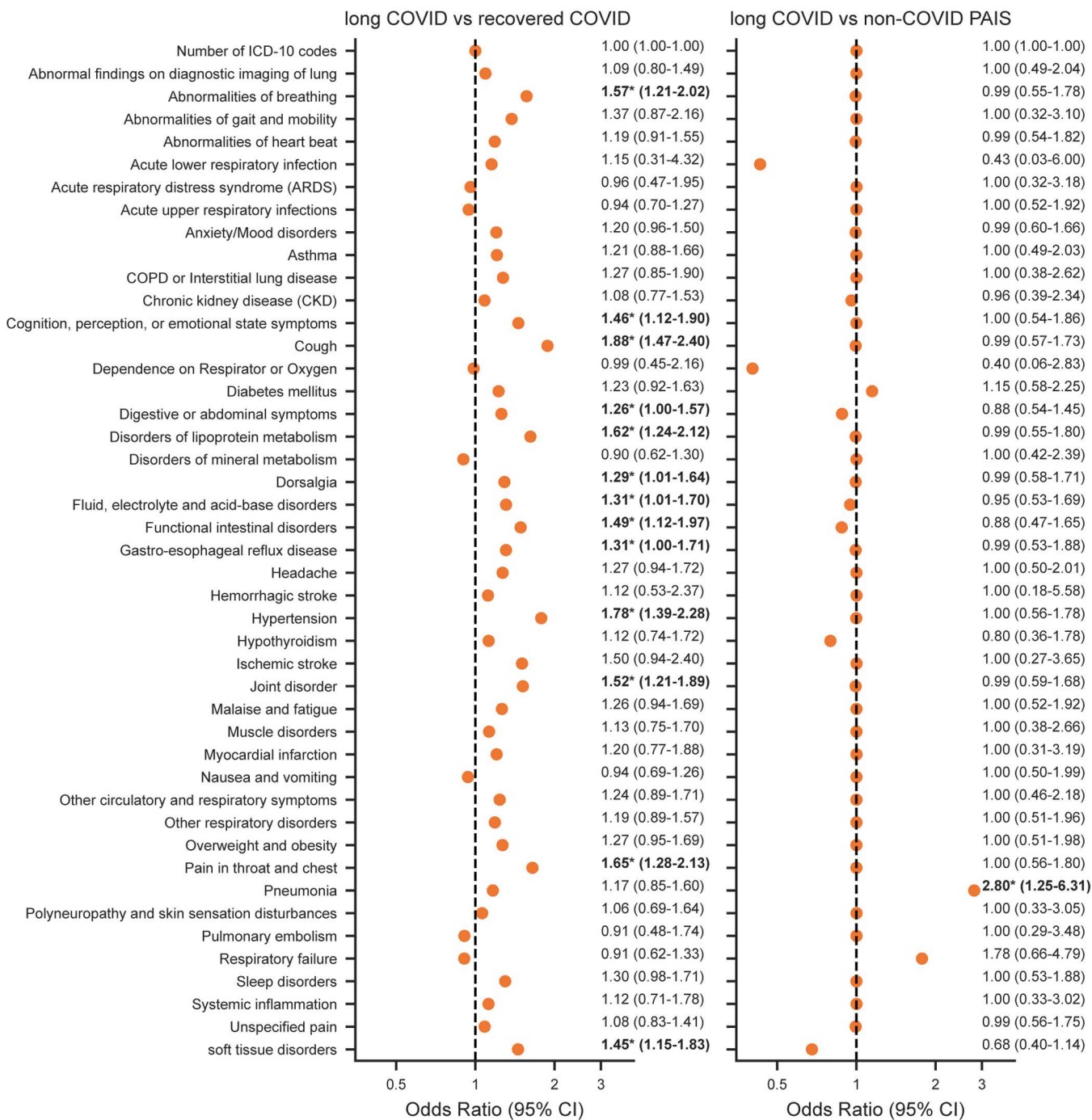

**Fig 3. Association of long COVID and symptoms/diagnoses prior to the index SARS-CoV-2 test.** Odds ratio of symptoms/diagnoses present prior to the index SARS-CoV-2 test for long COVID relative to recovered COVID (left) and non-COVID PAIS (right) calculated using a logistic regression model adjusted for age, sex, race/ethnicity, smoking exposure, and CCI scores. The odds ratio value and 95% confidence interval (CI) are shown to the right of each point. *Denotes p-values < 0.05.

**Table 1. Characteristics of the long COVID, recovered COVID, and non-COVID PAIS cohorts.**

| Characteristic | long COVID (n = 580) | recovered COVID (n = 7,437) | non-COVID PAIS (n = 106) | p value[b] | p value[c] |
|---|---|---|---|---|---|
| Age in years, median (IQR)[a] | 54 (39–66) | 46 (31–61) | 51 (34–63) | <0.001 | 0.12[a] |
| Charlson comorbidity index, median (IQR)[a] | 2 (0–4) | 1 (0–3) | 1 (0–3) | <0.001 | 0.03 |
| Follow-up time in days, median (IQR) [a] | 791 (322–980) | 278 (6–776) | 811 (452–984) | <0.001 | 0.28 |
| Self-Reported Sex, n (%) | | | | 0.87 | 0.22 |
| Male | 300 (51.7%) | 3,872 (52.1%) | 48 (45.3%) | | |
| Female | 280 (48.3%) | 3,563 (47.9%) | 58 (54.7%) | | |
| Other | 0 | 2 | 0 | | |
| Race/ Ethnicity, n (%) | | | | <0.001 | <0.001 |
| White | 266 (45.9%) | 3,070 (41.3%) | 70 (66.0%) | | |
| Hispanic | 116 (20.0%) | 1,253 (16.8%) | 11 (10.4%) | | |
| Black or African American | 99 (17.0%) | 1,146 (15.4%) | 6 (5.7%) | | |
| Asian | 48 (8.3%) | 664 (9.0%) | 12 (11.3%) | | |
| American Indian or Alaska Native | 11 (1.9%) | 103 (1.4%) | 3 (2.8%) | | |
| Native Hawaiian or Pacific Islander | 9 (1.6%) | 158 (2.1%) | 0 | | |
| Race Unknown | 31 (5.3%) | 1,043 (14.0%) | 4 (3.8%) | | |
| Smoking exposure, n (%) | | | | <0.001 | 0.77 |
| Yes | 167 (28.8%) | 1,267 (17.0%) | 32 (30.2%) | | |
| No | 413 (71.2%) | 6,170 (83.0%) | 74 (69.8%) | | |
| Hospitalized, n (%) | | | | <0.001 | <0.001 |
| Yes | 272 (46.9%) | 1,965 (26.4%) | 79 (74.5%) | | |
| No | 308 (53.1%) | 5,472 (73.6%) | 27 (25.5%) | | |

[a]IQR, interquartile range. [b]p values were calculated by comparing the long COVID and recovered COVID cohorts using two-sided Mann-Whitney $U$ tests for continuous variables and chi-squared tests for categorical data. [c]p values were calculated by comparing the long COVID and non-COVID PAIS cohorts using two-sided Mann-Whitney $U$ tests for continuous variables and chi-squared tests for categorical data.

While the long COVID patients were more similar to the recovered COVID group in terms of sex (51.7% and 52.1% male, $p = 0.87$) than to the non-COVID PAIS group (45.3% males, $p = 0.22$), there was more racial and ethnic diversity in both COVID groups compared to the non-COVID PAIS group (45.9%, 41.3% and 66.0% White race in the long COVID, recovered and non-COVID PAIS groups, respectively; and 20.0%, 16.8%, and 10.4% Hispanic, respectively, $p < 0.001$). The long COVID and non-COVID PAIS groups had similar smoking exposure (28.8% and 30.2%) which were higher than that in the recovered group (17.0%, $p < 0.001$ for long and recovered group comparison). The long COVID group had higher CCI scores (median 2.0, IQR 0–4) than either of the other two groups (median 1.0 and IQR 0–3 for both recovered and PAIS groups, $p < 0.05$ for both comparisons) although those in the non-COVID PAIS group were more likely to have been hospitalized within seven days of their index SARS-CoV-2 test (74.5% non-COVID PAIS versus 46.9% and 26.4% for the long and recovered COVID groups, $p < 0.001$ for both comparisons). The follow-up time distributions for the long COVID and non-COVID PAIS groups were similar [median (IQR), 791 (322–980) and 811 (452–984), $p = 0.28$] and were longer than the recovered COVID group [median (IQR), 278 (6–776), $p < 0.001$].

### Prior symptoms and diagnoses

Figs 3–5 graphically represent the differences and similarities between the long COVID group compared to the recovered COVID and non-COVID PAIS groups for having each diagnosis, based on the odds of the code in the long COVID versus each comparison group in the multivariable regression models. In the timeframe before the index SARS-CoV-2 test,

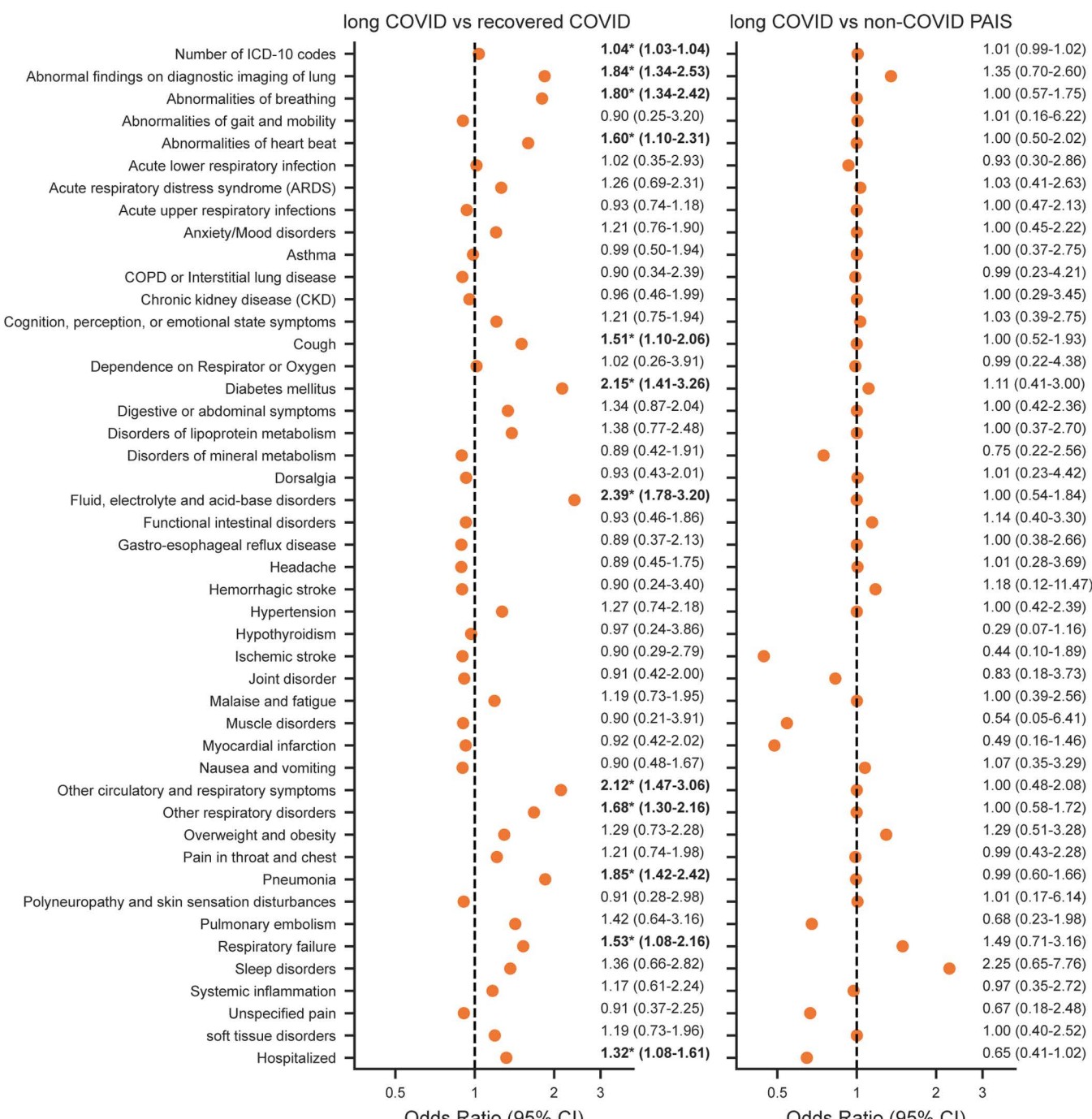

**Fig 4. Association of long COVID and symptoms/diagnoses within 30 days of the index SARS-CoV-2 test.** Odds ratio of symptoms/diagnoses present within the first 30 days of the index SARS-CoV-2 test for long COVID relative to recovered COVID (left) and non-COVID PAIS (right) calculated using a logistic regression model adjusted for age, sex, race/ethnicity, smoking exposure, and CCI scores. The odds ratio value and 95% confidence interval (CI) are shown to the right of each point. *Denotes p-values < 0.05.

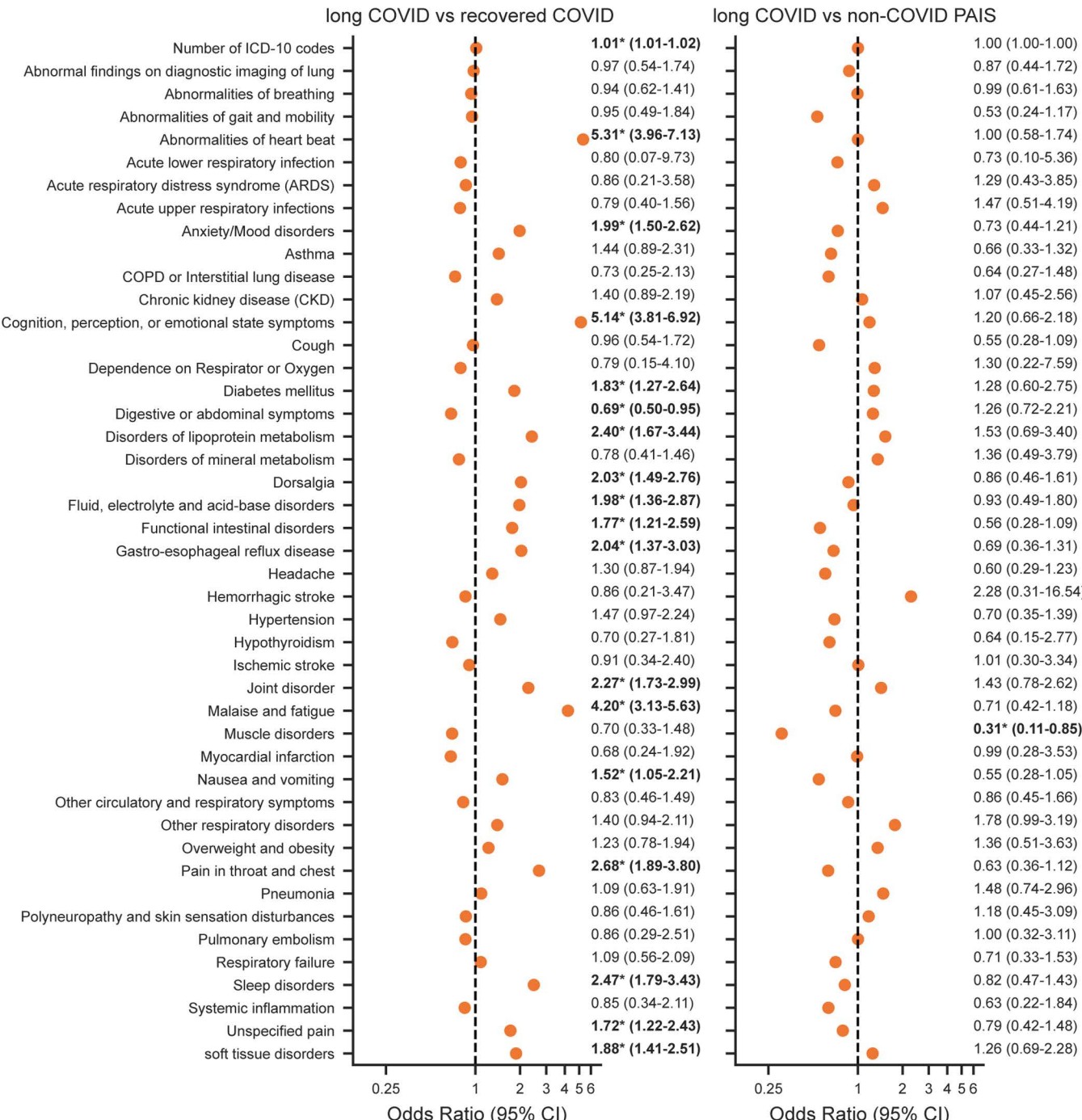

**Fig 5. Association of long COVID and symptoms/diagnoses present at least 30 days after the index SARS-CoV-2 test.** Odds ratio of symptoms/ diagnoses present at least 30 days after the index SARS-CoV-2 test for long COVID relative to recovered COVID (left) and non-COVID PAIS (right) calculated using a logistic regression model adjusted for age, sex, race/ethnicity, smoking exposure, and CCI scores. The odds ratio value and 95% confidence interval (CI) are shown to the right of each point. *Denotes *p*-values < 0.05.

pneumonia was the only diagnosis that was more likely to be documented among those who later developed long COVID than among the non-COVID PAIS group [OR (95% CI), 2.80 (1.25–6.31)] (Fig 3). All other diagnoses and symptoms were similarly present in the medical records of the long COVID and non-COVID PAIS groups before the index SARS-CoV-2 test. Compared to the recovered COVID group,13 symptoms and diagnoses were more likely to be documented in those who later developed long COVID group (Fig 3). The strongest associations included: cough [OR (95% CI), 1.88 (1.47–2.40); $p < 0.05$], hypertension [OR (95% CI), 1.78 (1.39–2.28); $p < 0.05$], pain in throat and chest [OR (95% CI), 1.65 (1.28–2.13); $p < 0.05$], disorders of lipoprotein metabolism [OR (95% CI), 1.62 (1.24–2.12); $p < 0.05$], and abnormalities of breathing [OR (95% CI), 1.57 (1.21–2.02); $p < 0.05$].

## Acute symptoms and diagnoses

During the timeframe within 30 days of the index SARS-CoV-2 test, no diagnostic codes were more likely to have been documented among those who later developed long COVID versus the non-COVID PAIS group (Fig 4). More than half (28/46) of the diagnosis groups had odds ratios close to 1, (although 95% confidence intervals were wide for many). However, compared with the recovered COVID cohort, people with long COVID were more likely to have codes for symptoms and diagnoses related to more severe infections: pneumonia [OR (95% CI), 1.85 (1.42–2.42); $p < 0.05$], abnormal findings on diagnostic imaging of lung [OR (95% CI), 1.84 (1.34–2.53); $p < 0.05$], and respiratory failure [OR (95% CI), 1.53 (1.08–2.16); $p < 0.05$] (Fig 4). Other positively associated features included metabolism-related conditions, such as fluid, electrolyte, and acid-base disorders [OR (95% CI), 2.39 (1.78–3.20); $p < 0.05$] and diabetes mellitus [OR (95% CI), 2.15 (1.41–3.26); $p < 0.05$]. In addition, the long COVID group was more likely to be hospitalized during this timeframe than the recovered COVID group [OR (95% CI), 1.32 (1.08–1.61)], but this association did not hold up for the comparison of the long COVID group with the non-COVID PAIS group [OR (95% CI), 0.65 (0.41–1.02)].

## Post-acute symptoms and diagnoses

The long COVID and non-COVID PAIS cohorts shared many similarities in the timeframe >30 days after the index SARS-CoV-2 PCR test (Fig 5). Of those symptoms/diagnoses with similar likelihood between the two groups, five had odds ratios between 0.99–1.01. Two groups represented general disorders: abnormalities of breathing [OR (95% CI), 0.99 (0.61–1.63)] and abnormalities of heartbeat [OR (95% CI), 1.00 (0.58–1.74)]. Three others were specific cardiopulmonary events: ischemic stroke [OR (95% CI), 1.01 (0.30–3.34)], myocardial infarction [OR (95% CI), 0.99 (0.28–3.53)], and pulmonary embolism [OR (95% CI), 1.00 (0.32–3.11)]. In contrast, many symptoms and diagnoses were more likely in the period after the index test among the long COVID group compared with the recovered COVID group. The strongest associations included: abnormalities of heartbeat [OR (95% CI), 5.31 (3.96–7.13); $p < 0.05$], cognition, perception, or emotional state symptoms [OR (95% CI), 5.14 (3.81–6.92); $p < 0.05$], malaise and fatigue [OR (95% CI), 4.20 (3.13–5.63); $p < 0.05$], pain in throat and chest [OR (95% CI), 2.68, (1.89–3.80); $p < 0.05$], and sleep disorders [OR (95% CI), 2.47, (1.79–3.43); $p < 0.05$]. These same symptoms/diagnoses were equally likely to be found in the long COVID and non-COVID PAIS patients. Only one diagnosis, muscle disorders, was found to be less common in non-COVID PAIS patients than in long COVID patients [OR (95% CI), 0.31 (0.11–0.85); $p < 0.05$]. All other symptoms/diagnosis were of approximately equal likelihood in both groups (although 95% confidence intervals were wide for many).

## Discussion

Using data extracted from patient medical records in a large community-based healthcare system, we evaluated demographic and clinical characteristics of patients with long COVID in comparison to individuals with non-COVID post-acute infectious syndromes (non-COVID PAIS) and patients who were SARS-CoV-2 PCR positive who did not develop long COVID (recovered COVID). These comparisons allowed us to identify common symptoms/diagnoses across the groups

and were intended to be hypothesis generating. We found associations of the symptoms and diagnoses with long COVID compared with non-COVID PAIS were similar across all timeframes (although power to detect differences was low). This was true even though the date of the triggering infection for the non-COVID PAIS cohort was unknown. Conversely, there were many symptoms and diagnoses that were more strongly associated with long COVID relative to recovered COVID for all three timeframes. Whereas the younger age of the recovered patients with COVID-19 may have affected this comparison, associations remained after adjusting for age.

Diagnoses present prior to the index test, such as hypertension and lipoprotein metabolism disorders, were more likely among the long COVID group versus the recovered COVID group. Similarly, during the timeframe within 30 days of the index SARS-CoV-2 test, features related to more severe infections were more likely in the patients with long COVID. These associations are consistent with previous studies highlighting the importance of pre-existing conditions and severity of acute infections for the development of long COVID [27–30,32]. Likewise, many of the organ systems affected by the symptoms and diagnoses identified in the period >30 days following the index SARS-CoV-2 test, such as cardiovascular [33], neurologic [34], and musculoskeletal [35], have been well documented to be associated with long COVID [3,23,36] versus recovery from COVID. The link between SARS-CoV-2 infections and diabetes has also been noted in previous studies [26,37–39].

Severity of illness within 30 days of the index SARS-CoV-2 test may also partially explain similarities that emerged between long COVID and non-COVID PAIS patients in the subsequent timeframe. Although the CCI was highest in the long COVID group, we found more hospitalizations within seven days of the index SARS-CoV-2 test for the non-COVID PAIS patients (74.5%) and long COVID patients (46.5%) than in the recovered COVID group (26.4%). Higher smoking exposure was also found in both the long COVID (28.8%) and non-COVID PAIS groups (30.2%) compared to the recovered COVID group (17.0%). As the literature on long term sequelae of post-acute infections is largely presented as disease-specific, research on post-acute infection syndromes as a group may be helpful to identify other commonalities to aid in identification and treatment of both long COVID and other similar syndromes.

The large numbers of people experiencing lingering symptoms of long COVID in the aftermath of the COVID-19 pandemic heightened awareness of PAIS in general, and evidence has been building that PAIS follow a wide variety of infections and may have a broader public health impact than was previously recognized. For example, Vivaldi et al.[19] compared long COVID with the long-term outcomes following non-specific acute respiratory infections and found there was little difference in symptoms or health-related quality of life measures. PAIS is likely underdiagnosed in clinical settings, particularly when the triggering infection is asymptomatic, or not identified. Studies comparing long COVID with long-term effects of other infections have generally defined non-COVID PAIS by the presence of long-term symptoms following a negative SARS-CoV-2 test. Misclassification is a concern for non-COVID PAIS comparison groups identified in this way. A strength of this study is that the non-COVID PAIS comparison group was identified by both a negative SARS-CoV-2 test and a documented diagnosis of PAIS, without indication of COVID-19 in the medical record. However, because of the challenges of diagnosing PAIS, use of diagnostic codes to identify the non-COVID comparison group may have introduced selection bias, e.g., a larger proportion of the non-COVID PAIS group was hospitalized within 30 days of the index SARS-CoV-2 test than among the long COVID or recovered COVID groups. We also cannot rule out misclassification of SARS-CoV-2 status. False positive and false negative SARS-CoV-2 test results were possible and we cannot exclude the possibility that some of the patients in the non-COVID PAIS group may have tested positive outside of the University of Washington network or had an undocumented SARS-CoV-2 infection.

Several other limitations should be considered when evaluating these results. Most important is the relatively small number of non-COVID PAIS patients (n = 106). Whereas identifying this group was an innovation of our study, this sample size provided limited power to detect differences between groups. However, in reporting results we attempted to look at effect size (odds ratio) in addition to the p-value for comparisons. A second limitation was the potential for individuals to have been misclassified as having long COVID because long COVID is not well-defined, and nonspecific ICD-10 codes

were used to identify patients with probable long COVID [40]. Thirdly, the infectious agent and date of infection was unknown for the non-COVID PAIS cohort, which reduced the ability to make inferences about the clinical comparisons to the long COVID group due to potential differences in length of illness. Fourthly, these analyses reflect one geographic location in the United States so generalizing to larger populations must be done with care. We also acknowledge that the time period used for classifying a patient with long COVID in this study was 30 days after the initial infection, which has recently been updated to a period of 3-months [4]. Whereas these results may not reflect sequelae of the new long COVID definition, they are still helpful for comparison to the literature on COVID-19 that was available prior to this change. Finally, we did not correct for multiple comparisons when reporting results as our analyses were intended to be hypothesis generating. Nonetheless, these findings add to the body of literature suggesting that further research comparing non-specific PAIS with long-term consequences of documented infections is warranted.

## Conclusion

The SARS-CoV-2 pandemic and associated long COVID cases have brought increased attention to non-COVID PAIS. This study helps describe the symptoms and diagnoses associated with long COVID relative to non-COVID PAIS and recovered patients with COVID-19 in different timeframes. The similarities between long COVID and non-COVID PAIS imply that factors related to persistent or new symptoms following some acute infectious diseases may be independent of the infectious agent. More studies are needed to distinguish the clinical characteristics that are shared versus those that are pathogen specific.

## Supporting information

**S1 Table. ICD-10 codes used to calculate Charlson comorbidity index scores.**
(PDF)

**S2 Table. ICD-10 codes used to categorize diagnoses and symptoms in Figs 3–5.**
(PDF)

## Acknowledgments

The authors would like to thank everyone on the COVID-RELIEF team for valuable advice and suggestions throughout the study as well as patients who provided data in the EHR. The flow diagram in Fig 1 was created with BioRender.com. The findings and conclusions in this report are those of the authors and do not necessarily represent the official position of the Centers for Disease Control and Prevention.

## Author contributions

**Conceptualization:** Caleb R. Carr, Nicole L. Gentile, Annette L Fitzpatrick.

**Data curation:** Annette L Fitzpatrick.

**Formal analysis:** Caleb R. Carr.

**Funding acquisition:** Annette L Fitzpatrick.

**Supervision:** Nicole L. Gentile, Annette L Fitzpatrick.

**Visualization:** Caleb R. Carr.

**Writing – original draft:** Caleb R. Carr.

**Writing – review & editing:** Caleb R. Carr, Nicole L. Gentile, Jeanne Bertolli, Warren Szewczyk, Jin-Mann S. Lin, Elizabeth R. Unger, Quan M. Vu, Nona Sotoodehnia, Annette L Fitzpatrick.

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
