## [Decision Letter · Decision Letter 0]

20 Dec 2024

PONE-D-24-52642Comparison of Long COVID, Recovered COVID, and Non-COVID Post-Acute Infection Syndromes over three yearsPLOS ONE

Dear Dr. Fitzpatrick,

Thank you for submitting your manuscript to PLOS ONE. After careful consideration, we feel that it has merit but does not fully meet PLOS ONE’s publication criteria as it currently stands. Therefore, we invite you to submit a revised version of the manuscript that addresses the points raised during the review process.

We look forward to receiving your revised manuscript.

Kind regards,

Dong Keon Yon, MD, FACAAI, FAAAAI

Academic Editor

PLOS ONE

Journal Requirements:

“Funding for this project was received through a contract from the Centers for

Disease Control and Prevention (CDC), contract number 75D30121C10207.”

6. We note that there is identifying data in the Supporting Information file <S1_File.pdf>. Due to the inclusion of these potentially identifying data, we have removed this file from your file inventory. Prior to sharing human research participant data, authors should consult with an ethics committee to ensure data are shared in accordance with participant consent and all applicable local laws.

-Location data

Additional Editor Comments:

Please address the excellent comments from the reviewers.

Reviewers' comments:

Reviewer's Responses to Questions

**Comments to the Author**

1. Is the manuscript technically sound, and do the data support the conclusions?

Reviewer #1: Yes

Reviewer #2: Yes

2. Has the statistical analysis been performed appropriately and rigorously? 

Reviewer #1: Yes

Reviewer #2: Yes

3. Have the authors made all data underlying the findings in their manuscript fully available?

Reviewer #1: No

Reviewer #2: Yes

4. Is the manuscript presented in an intelligible fashion and written in standard English?

Reviewer #1: Yes

Reviewer #2: Yes

5. Review Comments to the Author

Reviewer #1: - The article is well-written, easy to read, and interesting.

- For Non-COVID PAIS patients, it would be interesting to know if there is another associated infectious diagnosis (etiological or clinical) to better understand what other microorganisms might cause this post-viral syndrome. Considering there are only 106 patients, I am not sure if it would be feasible to determine whether there is a record of any infectious condition within the six months prior to the Non-COVID PAIS diagnosis.

- In Table 1, it might be helpful to record only "Yes" responses to simplify the table. For non-U.S. readers, discussing race and ethnicity separately can feel unusual; perhaps these could be categorized under a single category.

- I do not fully understand the extracted codes for symptoms and clinical history. When are the codes extracted, and from where? My understanding is that these codes pertain to a distinct episode of COVID/recovered/non-COVID and are not pulled from free text within the COVID/recovered/Non-COVID episode. It might be helpful to explain this in the methods section with a table: one column for the codes related to the patient’s medical history prior to the COVID/recovered/Non-COVID episode, another for the symptoms/abnormalities (e.g., pneumonia, abnormal radiological findings) recorded during the illness, and a final column for symptom codes that emerge afterward. When presenting the results, this structure should be followed. Mixing everything together makes the presentation less clear.

- Figures 3 and 4 are very interesting, but the data are mixed, making them harder to interpret. Categorizing them by personal history, clinical presentation, and complications could improve clarity. For example, it is unclear whether stroke or thromboembolism occurred after the episode or if they are pre-existing conditions. I assume they are complications, but the graph does not make this clear.

Reviewer #2: This study presents an interesting approach by comparing Long COVID with other post-acute infection syndromes (PAIS) to identify potential shared disease mechanisms. The findings, particularly the similarities between Long COVID and PAIS, are relevant and valuable for clinicians and researchers. However, I have several concerns about this study before publication.

1. The methods in the abstract should briefly describe the analytical techniques used in the study, such as multivariate regression or other statistical approaches used to derive results.

2. The methods in the abstract should specifically specify the total number of individuals included in the Washington University School of Medicine Electronic Health Records (EHR) database.

3. The abstract results should present the final sample size for each group (long COVID, recovered COVID, non-COVID PAIS) to clearly indicate the scale of the analysis; this information should not be left in the Methods section.

4. The definition of long-term COVID used in this study should be clarified. Because acute sequelae usually occur over a longer period of time, including data within 30 days of SARS-CoV-2 index testing seems too short to define long-term COVID.

5. The study does not mention how reinfection cases were handled. In order to accurately interpret the findings, it is important to describe whether individuals who have been infected multiple times with SARS-CoV-2 were included, excluded, or analyzed separately.

6. You should provide details on how the individual follow-up end date was determined. You should also report the mean and median of the follow-up periods for each group to clarify the temporal range of the study.

7. The definition of PAIS and its rationale should be specified accurately. The inclusion criteria of PAIS diagnostic codes should be justified to ensure the validity of the comparisons conducted in the study.

8. The definition and rationale for the analyzed results should be further explained. For example, how have heart rhythm abnormalities, cognitive and emotional symptoms, and sleep disorders been identified? The source or diagnostic criteria for these results should be clearly explained.

6. PLOS authors have the option to publish the peer review history of their article (what does this mean? ). If published, this will include your full peer review and any attached files.

**Do you want your identity to be public for this peer review?** For information about this choice, including consent withdrawal, please see our Privacy Policy .

Reviewer #1: No

Reviewer #2: No

---

## [Author Response · Author response to Decision Letter 1]

12 Feb 2025

RESPONSE TO EDITORS AND REVIEWERS’ COMMENTS:

We have checked for style and formatting requirements and believe that we have followed all of them .

There is no funding information in the manuscript.

“Funding for this project was received through a contract from the Centers for

Disease Control and Prevention (CDC), contract number 75D30121C10207.”

Role of funders is as follows:

The funder’s role was consultation on the analysis and interpretation of data, providing input on the decision to publish, and participation in the preparation of the manuscript.

Our data have ethical restrictions for sharing as the source is from electronic health records of patients in a healthcare system. Although medical record numbers have been removed, other data including dates and locations of care are included. It may be possible to identify individuals from one or a combination of data fields. Our IRB approval through the University of Washington does not allow our sharing of these records. Should someone wish to use our data, we can accommodate requests by addressing their individual needs. Data requests may be made to:

Dr. Kari Stephens

Vice Chair for Research

Department of Family Medicine

University of Washington, Seattle WA 98195

kstephen@uw.edu

NA

In Methods we include our IRB approval number for a waiver of consent.

6. We note that there is identifying data in the Supporting Information file <S1_File.pdf>. Due to the inclusion of these potentially identifying data, we have removed this file from your file inventory.

We are not sure why you believed our S1 supplementary file has identifying information in it as the tables only reflect groupings of ICD-10 codes used in the manuscript. We have uploaded it again and hope that you will include it for the reviewers as its omission created some confusion with one of the reviewers.

Additional Editor Comments:

Please address the excellent comments from the reviewers.

Reviewer #1: - The article is well-written, easy to read, and interesting.

We appreciate your general overview.

1) For Non-COVID PAIS patients, it would be interesting to know if there is another associated infectious diagnosis (etiological or clinical) to better understand what other microorganisms might .cause this post-viral syndrome. Considering there are only 106 patients, I am not sure if it would be feasible to determine whether there is a record of any infectious condition within the six months prior to the Non-COVID PAIS diagnosis.

This is a great question and for a very few individuals, an infectious organism (e.g., influenza, infectious gastroenteritis, or tuberculosis) could be identified as a probable cause. However, for most of the patients in our study, ICD-10 codes were not included to identify microorganisms. As a result, we could not make any inferences about other organisms that cause post-infectious syndromes, and this is stated as a limitation in the discussion.

2) In Table 1, it might be helpful to record only "Yes" responses to simplify the table. For non-U.S. readers, discussing race and ethnicity separately can feel unusual; perhaps these could be categorized under a single category.

We have revised Table 1 and grouped race and ethnicity together as suggested. We agree it is easier to understand.

3) I do not fully understand the extracted codes for symptoms and clinical history. When are the codes extracted, and from where? My understanding is that these codes pertain to a distinct episode of COVID/recovered/non-COVID and are not pulled from free text within the COVID/recovered/Non-COVID episode. It might be helpful to explain this in the methods section with a table: one column for the codes related to the patient’s medical history prior to the COVID/recovered/Non-COVID episode, another for the symptoms/abnormalities (e.g., pneumonia, abnormal radiological findings) recorded during the illness, and a final column for symptom codes that emerge afterward. When presenting the results, this structure should be followed. Mixing everything together makes the presentation less clear.

We completely agree with this comment. In the submitted manuscript we had structured the study – and have presented results – to show the different time periods in which the ICD-codes were included in the patient record. For each individual, the codes are grouped relative to the index PCR test: prior to the test, within 30 days, and greater than 30 days. These three timeframes correspond to Figures 3, 4, and 5. We have modified text in Methods to try to make this more clear.

4) Figures 3 and 4 are very interesting, but the data are mixed, making them harder to interpret. Categorizing them by personal history, clinical presentation, and complications could improve clarity. For example, it is unclear whether stroke or thromboembolism occurred after the episode or if they are pre-existing conditions. I assume they are complications, but the graph does not make this clear.

As mentioned above, Figures 3, 4, and 5 correspond to the three timeframes relative to the index PCR test: prior to the test, within 30 days, and greater than 30 days. The two different graphs in each figure represent the comparisons between the Long COVID group and the two other groups. We have tried to clarify this distinction in Methods and the Figure legends.

Reviewer #2: This study presents an interesting approach by comparing Long COVID with other post-acute infection syndromes (PAIS) to identify potential shared disease mechanisms. The findings, particularly the similarities between Long COVID and PAIS, are relevant and valuable for clinicians and researchers. However, I have several concerns about this study before publication.

Thank you.

1) The methods in the abstract should briefly describe the analytical techniques used in the study, such as multivariate regression or other statistical approaches used to derive results.

This is a great point and we now have adjusted the abstract to describe the use of multivariate regression to analyze the different groups in the three different timeframes.

2) The methods in the abstract should specifically specify the total number of individuals included in the Washington University School of Medicine Electronic Health Records (EHR) database.

We have adjusted the abstract to mention the total number (n=139,472) of individuals in the University of Washington Medicine electronic health record (EHR) database extracted for the study.

3) The abstract results should present the final sample size for each group (long COVID, recovered COVID, non-COVID PAIS) to clearly indicate the scale of the analysis; this information should not be left in the Methods section.

The sample size for each group was already included in the abstract but we have tried to make it more obvious by restructuring the sentence.

4) The definition of long-term COVID used in this study should be clarified. Because acute sequelae usually occur over a longer period of time, including data within 30 days of SARS-CoV-2 index testing seems too short to define long-term COVID.

We defined a probable Long COVID group from the SARS-CoV-2 PCR positive base population by selecting individuals with an ICD-10-CM diagnostic code of U09.9 (post COVID-19 condition, unspecified), B94.8 (sequelae of other specified infectious and parasitic diseases), and/or G93.3 (postviral fatigue syndrome) at least 30 days after their positive index SARS-CoV-2 PCR test. We used 30 days from the index PCR test date as the cut point to define the post-acute infection period because previous literature indicated many symptoms associated with post-COVID-19 conditions persist or begin four weeks after the initial SARS-CoV-2 infection and early Long COVID definitions used during the time of data collection utilized a four week post-infection cutoff. We acknowledge that the current definition of Long COVID has been revised to 3 months or longer post-infection (National Academy of Sciences, Engineering and Medicine 2024), however, physicians providing codes at the time these data were collected not have considered this definition.

5) The study does not mention how reinfection cases were handled. In order to accurately interpret the findings, it is important to describe whether individuals who have been infected multiple times with SARS-CoV-2 were included, excluded, or analyzed separately.

The outcome in this study is defined as the first incidence of a positive SARS-COV-2 PCR test. For those with more than one positive test, the first positive was defined as the index infection. This is noted in Methods.

6) You should provide details on how the individual follow-up end date was determined. You should also report the mean and median of the follow-up periods for each group to clarify the temporal range of the study.

The end date for this study was the last date a patient record was accessed (January 31, 2022) and is based on when data were downloaded from the EHR. A new row to Table 1 has been added that shows the median and interquartile range in days for the follow-up time for each group. An image of the new row is pasted below.

7) The definition of PAIS and its rationale should be specified accurately. The inclusion criteria of PAIS diagnostic codes should be justified to ensure the validity of the comparisons conducted in the study.

We defined PAIS based on ICD-10 codes found in patient records. Our definitions, included in the text, are as follows:

Long COVID group definition from text:

A probable Long COVID group was extracted from the SARS-CoV-2 PCR positive base population by selecting individuals with an ICD-10-CM diagnostic code of U09.9 (post COVID-19 condition, unspecified), B94.8 (sequelae of other specified infectious and parasitic diseases), and/or G93.3 (postviral fatigue syndrome) at least 30 days after their positive index SARS-CoV-2 PCR test. Because the U09.9 code was not introduced until October 1st, 2021, patients who received a U07.1 (COVID-19) code through September 30, 2021, at a healthcare encounter both prior to and 30 days after their positive index PCR test were included in this Long COVID group (n=580) (Fig 1). These ICD-10-CM codes were selected based on the recommendations of clinicians working in a Long COVID clinic at the University of Washington.

Non-COVID PAIS group definition from text:

A third group was identified from the SARS-CoV-2-PCR negative group who had no indication of COVID-19 but had a post infectious syndrome ICD-10-CM code (B94.8, sequelae of other specified infectious and parasitic diseases; G93.3: postviral fatigue syndrome) (n=106, Fig 1). No indication of COVID-19 was defined by having only negative PCR tests on record and no COVID-19-associated ICD-10-CM codes (U07.1: COVID-19; Z86.16: personal history of COVID-19; J12.82: pneumonia due to coronavirus disease 2019; U09.9: post COVID-19 condition, unspecified; B97.21: SARS-associated coronavirus as the cause of diseases classified elsewhere; B97.29: other coronavirus as the cause of diseases classified elsewhere; B34.2: coronavirus infection, unspecified).

8) The definition and rationale for the analyzed results should be further explained. For example, how have heart rhythm abnormalities, cognitive and emotional symptoms, and sleep disorders been identified? The source or diagnostic criteria for these results should be clearly explained.

We have included a supplement to the manuscript that provides all ICD-10 codes used in our definitions. We hope this will provide sufficient details on categories used in the paper.

---

## [Decision Letter · Decision Letter 1]

20 Mar 2025

PONE-D-24-52642R1Comparison of Long COVID, Recovered COVID, and Non-COVID Post-Acute Infection Syndromes over three yearsPLOS ONE

Dear Dr. Fitzpatrick,

Thank you for submitting your manuscript to PLOS ONE. After careful consideration, we feel that it has merit but does not fully meet PLOS ONE’s publication criteria as it currently stands. Therefore, we invite you to submit a revised version of the manuscript that addresses the points raised during the review process. Please submit your revised manuscript by May 04 2025 11:59PM. If you will need more time than this to complete your revisions, please reply to this message or contact the journal office at plosone@plos.org . Please include the following items when submitting your revised manuscript:

We look forward to receiving your revised manuscript.

Kind regards,

Dong Keon Yon, MD, FACAAI, FAAAAI

Academic Editor

PLOS ONE

Journal Requirements:

Additional Editor Comments:

This is an excellent paper. Finally, please see my comments.

#1. Long COVID -> long COVID

#2. Recovered COVID patients -> "r"ecovered patients with COVID-19

- Please use the patient-first language.

#3. Non-COVID -> "n"on-COVID

#4. Charlson Comorbidity Index -> Charlson comorbidity index

#5. In data section, the authors describe " smoking exposure (current, former, none", but in Table 1, yes or no.

#6. Likewise, many of the organ systems affected by the symptoms and diagnoses identified in the period >30 days following the index SARS-CoV-2 test, such as cardiovascular [#1], neurologic [#2], and musculoskeletal [#3], have been well documented to be associated with Long COVID [3, 32, 33]

Please add the top-tier paper reference. [1-3]

Reviewers' comments:

Reviewer's Responses to Questions

**Comments to the Author**

1. If the authors have adequately addressed your comments raised in a previous round of review and you feel that this manuscript is now acceptable for publication, you may indicate that here to bypass the “Comments to the Author” section, enter your conflict of interest statement in the “Confidential to Editor” section, and submit your "Accept" recommendation.

Reviewer #1: All comments have been addressed

2. Is the manuscript technically sound, and do the data support the conclusions?

Reviewer #1: Yes

3. Has the statistical analysis been performed appropriately and rigorously? 

Reviewer #1: Yes

4. Have the authors made all data underlying the findings in their manuscript fully available?

Reviewer #1: Yes

5. Is the manuscript presented in an intelligible fashion and written in standard English?

Reviewer #1: Yes

6. Review Comments to the Author

Reviewer #1: Thanks to the authors for their work and for addressing all my concerns. I believe the article should be accepted.

7. PLOS authors have the option to publish the peer review history of their article (what does this mean? ). If published, this will include your full peer review and any attached files.

**Do you want your identity to be public for this peer review?** For information about this choice, including consent withdrawal, please see our Privacy Policy .

Reviewer #1: No

---

## [Author Response · Author response to Decision Letter 2]

31 Mar 2025

Journal Requirements:

The reference list has been checked for completion and correctness. The only change was the addition of three references in relation to point #6 below.

Additional Editor Comments:

This is an excellent paper. Finally, please see my comments.

#1. Long COVID -> long COVID

This change has been implemented throughout the text and figures.

#2. Recovered COVID patients -> "r"ecovered patients with COVID-19

- Please use the patient-first language.

This change has been implemented throughout the text and figures.

#3. Non-COVID -> "n"on-COVID

This change has been implemented throughout the text and figures.

#4. Charlson Comorbidity Index -> Charlson comorbidity index

This change has been implemented throughout the text and figures.

#5. In data section, the authors describe " smoking exposure (current, former, none", but in Table 1, yes or no.

Current and former smokers were grouped as yes for smoking exposure, while “none” smokers were marked as no. The text in the data section has been modified to clearly state this classification

#6. Likewise, many of the organ systems affected by the symptoms and diagnoses identified in the period >30 days following the index SARS-CoV-2 test, such as cardiovascular [#1], neurologic [#2], and musculoskeletal [#3], have been well documented to be associated with Long COVID [3, 32, 33]

Please add the top-tier paper reference. [1-3]

The text and references have been modified to include references specifically for cardiovascular, neurologic, and musculoskeletal related issues, which are referenced in the manner specified above.

---

## [Editor Report · Decision Letter 2]

2 Apr 2025

Comparison of Long COVID, Recovered COVID, and Non-COVID Post-Acute Infection Syndromes over three years

PONE-D-24-52642R2

Dear Dr. Fitzpatrick,

We’re pleased to inform you that your manuscript has been judged scientifically suitable for publication and will be formally accepted for publication once it meets all outstanding technical requirements.

Kind regards,

Dong Keon Yon, MD, FACAAI, FAAAAI

Academic Editor

PLOS ONE

Additional Editor Comments (optional):

This is an excellent paper.
---

## [Editor Report · Acceptance letter]

PONE-D-24-52642R2

PLOS ONE

Dear Dr. Fitzpatrick,

I'm pleased to inform you that your manuscript has been deemed suitable for publication in PLOS ONE. Congratulations! Your manuscript is now being handed over to our production team.

Kind regards,

on behalf of

Dr. Dong Keon Yon

Academic Editor

PLOS ONE